# Effect of Different Water Salinities on the Larvae of the Blue Bream *Ballerus ballerus* (Linnaeus, 1758) during Rearing

**DOI:** 10.3390/ani13071245

**Published:** 2023-04-03

**Authors:** Przemysław Piech, Roman Kujawa

**Affiliations:** Department of Ichthyology and Aquaculture, Faculty of Animal Bioengineering, University of Warmia and Mazury in Olsztyn, 10-719 Olsztyn, Poland

**Keywords:** blue bream, rheophilic fish, salinity, larvae, survivability, growth rate

## Abstract

**Simple Summary:**

Climate change and negative anthropogenic pressures are becoming huge challenges for quite a number of populations of both terrestrial and aquatic animals. To a significant degree, fish—the most numerous representatives of vertebrates in the world—are exposed to changing temperature conditions but also to the physical and chemical conditions of inland waters. Any biotic or abiotic factor, to a greater or lesser extent, affects the adaptability of individual fish species to the prevailing conditions in the waters in a given area.

**Abstract:**

The influence of water salinities of 3, 5, and 7 ppt on the growth and survival of *Ballerus ballerus* (L.) larvae was studied. The control group was fish reared in freshwater (0 ppt). The larvae showed high tolerance to water salinities of 5–7 ppt. The mean final weight of the larvae ranged from 48.6 to 64.1 mg, with corresponding mean total lengths from 18.9 to 22.6 mm, depending on the water salinity level. The best larval length increments were recorded in water with salinity of 3 ppt. They were only slightly lower in 0 ppt water, and there were no statistically significant differences between the breeding rates calculated for larvae reared in 3 ppt water. Depending on the salinity level of the water, the final survival rate of the blue bream larvae ranged from 83.5 to 98.6%. The blue bream larvae reared in water with salinity levels of 5 and 7 ppt were statistically smaller than the others, but the results obtained were equally satisfactory.

## 1. Introduction

The quality of the environment, and especially of inland waters, is deteriorating continuously. These changes are so severe that they threaten many organisms [1]. There are many natural and anthropogenic factors that contribute to this state of affairs. Climate change, which in a way is a natural factor, can cause droughts (low water levels), as well as intense rainfall, resulting in catastrophic water surges [2,3]. In addition to the natural causes of negative processes in the terrestrial environment and in aquatic ecosystems, there are also those caused by irrational and ill-considered human activity. Rivers are under tremendous pressure due to artificial modifications of their flow, the discharge of domestic and industrial wastewater, and changes in catchment land use (such as agriculture and mining) [4,5,6]. These factors contribute in various ways to physical and chemical changes in water, including its salinity. Most often, chemical changes are due to the presence of substances that are not normally found in such quantities in watercourses, even in estuarine zones. Their impact on the organisms inhabiting a particular section of the watercourse is ambiguous. The negative effects of human-made developments interfering with the environment are not always immediately apparent. All mines, both open-pit and underground ones, developed for the extraction of various raw materials, pollute the environment with post-mining waters [7,8]. Post-mining waters originating from aquifers occur in every mine. They vary in their content of chloride and sulphide ions as well as heavy metal compounds. These waters, which contain large amounts of salts and heavy metals, end up in rivers and then in the seas. The inflow of post-mining waters into rivers causes their salinization, which is not without significance for river-dwelling organisms. It is worth noting at this point that saline waters originate from other sources as well. A periodic source of highly saline waters, occurring only in winter, is runoffs from roads [9]. This is associated with the use of salt as a means of reducing ice on roads. The most common anti-icing agents are inorganic-chloride-based salts, i.e., sodium chloride (NaCl), magnesium chloride (MgCl_2_), and calcium chloride (CaCl_2_). Once applied, road salts dissolve and enter freshwater ecosystems through saline overland flow generated by melting snow or rain and through groundwater sources [10,11].

The salinity of inland waters affects the biocenosis of inland waters, and especially the behaviour of aquatic organisms, including fish. Saline pollution creates a permanent chemical impact on aquatic biocenosis [12]. This is especially true for organisms with low mobility and those that are sensitive to changes in water salinity, such as juvenile stages of fish, molluscs, benthic crustaceans, and insect larvae [13,14,15,16]. The runoff of saline water with an inert composition into watercourses can lead to the extinction of sensitive populations of fish and other organisms living in a particular section of a river [17]. The permanent runoff of saline water and the inability of aquatic organisms to move to avoid the negative impact of water with a significant degree of salinity not only annihilates weaker and less resistant species, but also limits the development of less sensitive species. This is especially true for the early developmental stages of stenohaline cyprinid fish, which are hardly resistant to salinity changes [18].

Among the fish that do not avoid brackish waters are those in the so-called transitional waters. These fish benefit from food-rich marine waters, where they feed, and reproductively suitable freshwater, where they breed [19,20,21,22]. The early life stages of the listed fish species are much more tolerant to salinity than stenohaline fish. In estuarine zones, there are also species that very rarely enter saline waters and spend most of their lives in freshwater. An example is the blue bream *Ballerus ballerus* [23]. The blue bream is a rheophilic, social species of fish from the *Cyprinidae* family, occurring naturally in the Szczecin Lagoon, as well as Dąbie Lake, a delta lake in the Lower Odra River Valley. Despite the rather sizable population of this species in the aforementioned inland waters, there is a declining trend of “genetically pure populations” due to interspecies crossings. The most frequently observed are crossbreedings of the spud with the bream. These crossings most often occur in places where their spawning dates overlap [24]. Despite the fact that the blue bream is of minor economic importance, it is a component of the ichthyofauna that defines the biodiversity of rivers and lakes. In order to preserve biodiversity in inland waters, active and passive protection is extended to fish species of minor economic importance. A serious contribution to the active protection of species is provided by aquaculture. It is becoming commonplace to develop mechanisms to naturally support many economically valuable species. They range from species acclimation and reproductive biotechnology to the development of rearing protocols and rearing itself.

In the case of the blue bream, an effective method for its reproduction and optimal feeding during rearing under controlled conditions has already been developed [25,26]. Research on the effects of various factors on the growth and survival of blue bream larvae has been launched due to the lack of relevant data in the literature. A study was begun by determining the effect of salinity on the growth and survival of larvae. This is an issue of great interest in the context of the increasingly reported elevated salt levels in watercourses where this fish species still occurs [27]. 

In this study, a study of the effect of different degrees of water salinity was conducted. The study aimed to determine the appropriate level of water salinity at which fish achieve satisfactory growth rates and larval survival.

## 2. Materials and Methods

Blue bream spawners were caught in mid-May during their natural reproduction period from Lake Dąbie. The fish were caught using pond equipment—fyke nets and wontons. The blue bream larvae were from reproduction carried out under controlled conditions [25]. 

Larval rearing was carried out in four separate recirculating aquaculture systems (RASs) adapted to rearing larvae in water with different salinities. Blue bream larvae were reared in water with a salinity of 3, 5, or 7 ppt. The control sample consisted of larvae reared in freshwater (0 ppt), free of salinity. Pro-Reef sea salt (TROPIC MARIN, Wartenberg, Germany), which was dissolved in water at 20 °C, was used to achieve adequate water salinity. A single experimental circuit consisted of 3 aquaria designed for rearing larvae in water of a given salinity. Each aquarium had a working volume of 25 dm^3^. Each experimental circuit also included a 200 dm^3^ ‘filter’ aquarium containing a protein skimmer (EHEIM GmbH & Co.KG, Deizisau, Germany) and a biological bed in the form of a floating medium. The filter medium consisted of bactoballs used in denitrification filters (AB Aqua Medic GmbH, Bissendorf, Germany). The water circulation in each circuit and the water supply to the individual aquaria were identical to the methodology described by Kujawa and Piech when studying the effects of salinity on asp larvae (Figure 1) [28]. 

The blue bream larvae at the age of 5 DPH (days post hatching), with an average body length of 7.5 mm, were counted and placed in all rearing aquaria at a density of 40 indiv.·dm^−3^. The stocking density of a single aquarium was 1000 larvae. The water temperature in all the test groups was initially 20 °C and was gradually (1 °C per hour) increased to 25.0 ± 0.5 °C, which was the target rearing temperature. Measurements in each experimental group (salinity) were carried out in triplicate.

Control measurements of the weight and total length of the larvae were carried out every four days (day 1, 5, 9, 13, 17, and 21 of rearing). Each time, the samples consisted of thirty individuals (n = 30) randomly caught from each aquarium. They were caught in the morning (before feeding began). Fish were weighed to the nearest 0.1 milligram and measured to the nearest 0.1 mm [28].

The blue bream larvae were fed ad libitum three times a day at four-hour intervals (8 a.m., 12 p.m., and 4 p.m.). The food consisted of live nauplius of the brine shrimp *Artemia* sp. (Ocean Nutrition Europe, Essen, Belgium) [29]. 

The water oxygen content was measured with a YSI multiparameter meter (YSI Incorporated Brannum Lane, Yellow Springs, OH, USA) and was within 7.1–7.4 mg O_2_ dm^−^^3^, and the water pH range was 7.1–7.3. The ammonia and nitrite contents were measured in the morning with a HANNA INS HI 83200 with reagents (Hanna Instruments, Woonsocket, RI, USA), and neither were detected in the water during the experiment [30]. 

During the experiment, the number of dead larvae was counted daily, and the final survival rate was calculated in %. This was expressed as the ratio of the number of fish at the end of rearing to the number of larvae at the start of the experiment. Cumulative mortality curves were plotted for each rearing mode based on daily counts of dead larvae. The average PM larval weight gain (mg) was calculated for larvae from each experimental group studied at the end of the three-week rearing period [31]. Measurements were used to calculate the survival, the index of total length gain per unit time (ITL), the relative body growth rate (SGR), and the survival and larval biomass (SBR) [32,33]. Based on the above, the relative growth rate (RGR) and relative biomass growth rate (RBR) from the beginning of feeding to the end of the experiment were calculated [34]. Growth rates (SGR and RGR) for body length were calculated in the same way.

The biomass of fish in the tanks was determined as the product of the average weight of individuals and the number of live individuals. The obtained value was divided by the working volume of the tanks, obtaining the biomass of larvae expressed in g·dm^−3^. In order to compare the results obtained, the relative final average length, weight, and biomass of the experimental fish were calculated assuming that the length, weight, and biomass of fish from the control sample (from 0 ppt water) at the end of the experiment were equal to 100%. The assessment of the normality of the data distribution and the statistical processing of the results were carried out using the Excel 2016 and Statistica 13.0 for Windows (StatSoft, Inc., USA) computer programs. Statistical differences between the experimental groups were determined using Duncan’s test at a significance level of α = 0.05 [35].

## 3. Results

After adjusting the salinity of the water in the individual aquaria to the desired level, the first dead larvae were observed on the second day in water with a salinity of 7 ppt. In water with a salinity of 3 and 5 ppt, the first dead larvae were observed on the third day. During the 21-day rearing period, the lowest mortality was observed in the control water (0 ppt) (Figure 2).

Nearly 98.6% of individuals from the initial stock survived the experiment. A very similar survival rate of larvae was obtained when they were reared in water with salinity of 3 ppt. In this variant, 97% of the larvae survived the entire rearing period. The lowest survival rate of larvae was observed in water with salinity of 7 ppt, where 83.2% of larvae from the initial stock survived rearing (Table 1).

During all the control measurements, the average lengths of larvae reared in freshwater and in water with salinity of 3 ppt were similar (Figure 3).

On the last day of rearing, the blue bream larvae reared in 3 ppt water gained an average body weight of 64.1 mg with an average length of 22.6 mm (Figure 3 and Figure 4).

The smallest larvae were obtained by rearing them in water with salinity of 7 ppt. On the last day of rearing, they obtained an average weight of 48.6 mg, with an average length of 19.0 mm, while in water with salinity of 5 ppt, they obtained an average weight of 52.4 mg, with an average length of 19.2 mm. No statistically significant differences were noted between the average weight of individuals raised in salinity-free water and in water with salinity of 3 ppt. Similarly, no statistically significant differences were noted between the average weight of individuals reared in water with salinity of 5 and 7 ppt. The highest biomass of the blue bream larvae of 62.2 g (2.5 g·dm^−3^) was obtained in water with salinity of 3 ppt. The biomass of the blue bream larvae calculated for larvae from freshwater was only slightly lower, at 61.6 g (2.5 g·dm^−3^). These differences were not statistically significant. The lowest biomass of larvae was obtained in water with salinity of 7 ppt, in which it was only 40.4 g (1.6 g·dm^−3^). Table 1 also shows the results of the other studied breeding indicators such RGR and ITL. 

The calculations in Table 2 also confirm that the best rearing conditions for blue bream larvae were recorded in water with salinity of 3 ppt.

## 4. Discussion

Salinity is the total amount of dissolved mineral salts in water. It is determined by the absolute sum of salt concentrations in mg·dm^−^^3^ or in per mille. It can range from 2 to 390 mg·dm^−^^3^. Sources of salinity in watercourses can be runoff from agricultural fields, rising sea levels, the regulation of river hydrology, mining, and the use of salt to de-ice roads [36,37]. It affects all types of freshwater ecosystems—lakes, rivers, streams, and wetlands. It is one of many environmental factors inhibiting the establishment and spread of hydrobionts. It is a limiting factor for habitats or areas in which the success of a species’ reproduction is possible [13,38]. It affects the survival, metabolism, and occurrence of many fish species, which, as the most abundant group of hydrobionts vertebrates today, have mastered both low-ion-content waters—freshwaters—as well as those with high ion content—saline waters.

Research on the effects of different salinities on different life stages of freshwater fish has been conducted for a long time. Most of the studies have been on fish reared for consumption, in which the optimum salinity level to increase rearing efficiency in recirculating RAS systems has been determined [39,40,41,42,43,44]. Examples include studies on the rearing of larvae of sichel *Pelecus cultratus* [45] and asp *Leuciscus aspius* [28]. The above studies made it possible to raise more fish with less labour and energy. 

Current investigations in many research centres focus on determining the ability of various taxa to survive in water where salinity has occurred rather than on increasing the effects of rearing using saline water. The described experiment on blue bream larvae fits perfectly into this issue, as developing embryos and early-stage fish larvae, in addition to fish gametes, are particularly sensitive to changes in water salinity [13,14,15,16,28,45,46]. Freshwater fish are exposed to saline water not only when they are in the intertidal zone or when migrating from fresh to saline waters, but also while in their natural habitat. These are not, however, places on the borderline between fresh and saline waters, but sections of rivers heavily polluted with salts [10]. In fact, for several decades, there has been an increasing level of salinity in large sections of inland waters created by anthropogenic activities, such as mining, metallurgy, and chemical industries, and as a result of the runoff of highly saline water from stormwater networks during the winter [1,47,48]. Point or sectional salinity is not without influence on the biocoenoses of local water bodies, including the resident ichthyofauna [1,10,13]. This study confirms that blue bream larvae, like asp larvae inhabiting similar environments, also tolerate waters with salinity levels of 3–7 ppt [28]. Older individuals of rheophilic cyprinid fish are able to tolerate much higher water salinities. An example is the closely related asp *Leuciscus idus*, which tolerates water even with salinity of 15 ppt. Most likely, stenohaline organisms, which exhibit anadromous behaviour, inhabit mainly rivers and are often observed in waters with salinities of several per mille, will be the best adapted to these changes [49].

Differences in the growth of blue bream larvae in water with different salinity levels are confirmed by literature data, which show that changes in salinity result in greater energy consumption for osmotic and ionic regulation, meaning that less energy remains for growth and development [33,50,51,52]. Freshwater fish larvae placed in salt water (a hypertonic solution relative to body fluids) lose the water found in the tissues, resulting in an increase in salt concentration in the body. The acid–base balance of the blood is then violated. Haemoglobin is unable to properly bind oxygen found in the blood, which can lead to impaired oxygen metabolism in the body, and ultimately, death [53]. The above changes apply not only to larvae, but also to later life stages. The successful transfer of freshwater fish to high-salinity water involves the tolerance of tissues to changes in their hydration or the ability to regulate osmotic pressure, such as through the kidneys and gills. In the experiment conducted, it was observed that a gradual increase in salinity did not adversely affect the larvae of the blue bream. Small losses of larvae in the first days of the experiment were recorded in all aquaria with both salted and salt-free water.

The blue bream larvae had high survival and growth rates in water with salinity of 3 ppt, or 3 g·dm^−^^3^. According to a study by Lax and Peterson, sustained exposure to chloride concentrations above 25 mg·dm^−^^3^ can negatively affect freshwater fish [54]. The high tolerance of blue bream larvae to water with a salinity of 3 ppt indicates that there are large interspecies differences even among stenohaline carp in salt sensitivity [31]. The adaptability of fish to saline water may depend on previous salt exposure. Moreover, the mechanisms behind the possible acclimation or adaptation to salinity by freshwater organisms are still unexplored [55], and it is not known why some individuals of the same species better tolerate and others very poorly tolerate saline water [56,57].

The biological effects of salinity include the continued replacement of salt-intolerant organisms with those that can withstand elevated salt concentrations [58]. Increased salinity kills freshwater species due to toxic levels of sodium and chloride ions in their cells and the reduced ability to take up essential ions and water. These effects can reduce species diversity and significantly alter trophic systems by reducing food sources for consumers [59].

When discussing the effects of saline water on freshwater organisms, worth mentioning are the elements and chemical compounds that can potentiate the negative effects of chlorides and sulphides on organisms. Such elements are calcium, Ca, magnesium, Mg, and potassium, K. Sources of Ca and Mg can be road salts, such as CaCl_2_ and MgCl_2_, which are not only used in winter, but also in summer to bind dust on gravel roads [12]. The first two often form CaCO_3_ and MgCO_3_ compounds responsible for water hardness. The more of these compounds there are, the harder the water is, and the impact of chloride ions on organisms is lower [60]. In the described experiment, tap water (with a hardness of 24°n) was used as a diluent for sea salt. Presumably, this could have influenced such a high survival rate of blue bream larvae in all the salinities tested. The negative effects of chlorides and sulphides on freshwater organisms are compounded by the elevated presence of K [48]. Therefore, when rearing freshwater fish larvae in water of different salinity levels, as well as when predicting the ecological effects of introducing brine into inland waters, it is very important to consider the local water chemistry and the characteristics of the local species assemblage [61].

## 5. Conclusions

In the above experiment, the best growth parameters were obtained by rearing blue bream larvae in water with a salinity of 3 ppt. The higher salinity levels tested were also accepted by the blue bream larvae, as evidenced by the calculated rearing parameters. The survival rates of blue bream larvae in all salinities tested were very high, ranging from 83.2% to 97.0%. The tolerance of blue bream larvae to water with salinities of 3–7 ppt suggests that they are able to survive and thrive in natural waters with negligible salinities (up to 7 ppt).

## Figures and Tables

**Figure 1 animals-13-01245-f001:**
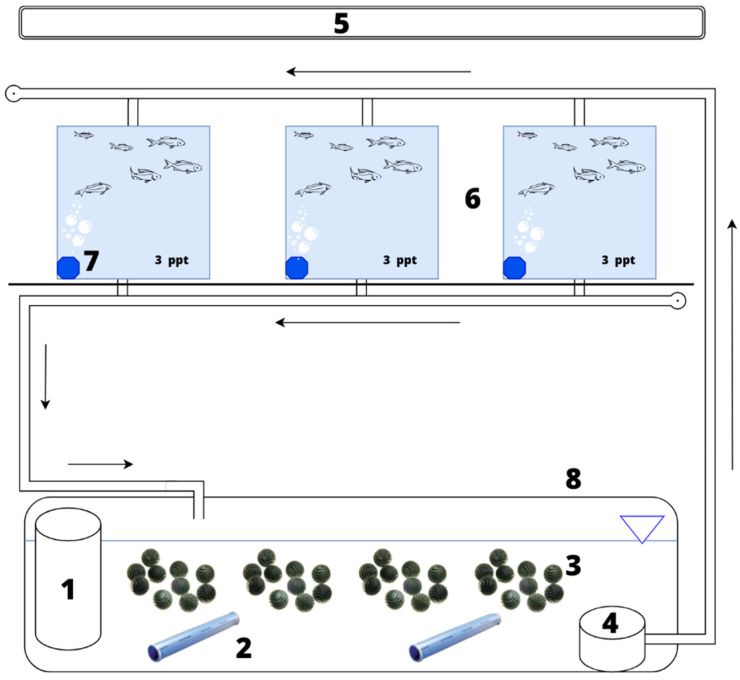
An overview diagram of one of the four prepared experimental systems used in the experiment: 1—protein skimmer, 2—tubular diffuser, 3—bactoballs, 4—pump, 5—fluorescent lamps, 6—aquariums with fishes, 7—aeration stone, 8—aquarium with filter bed.

**Figure 2 animals-13-01245-f002:**
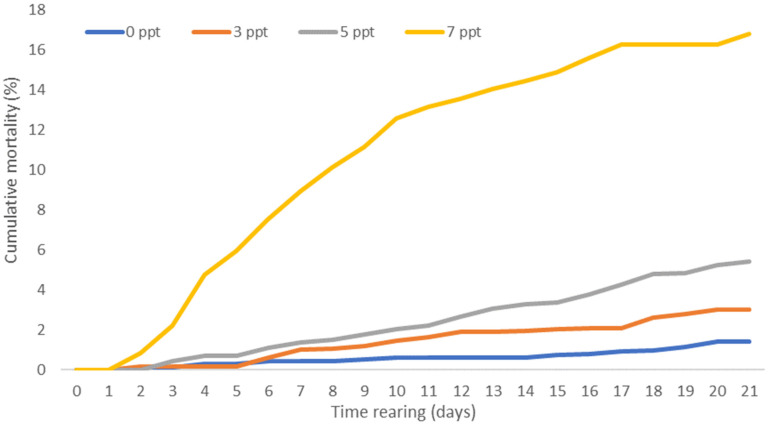
Cumulative mortality of blue bream *Ballerus ballerus* (L.) larvae reared in freshwater and water with salinity of 3, 5, and 7 ppt.

**Figure 3 animals-13-01245-f003:**
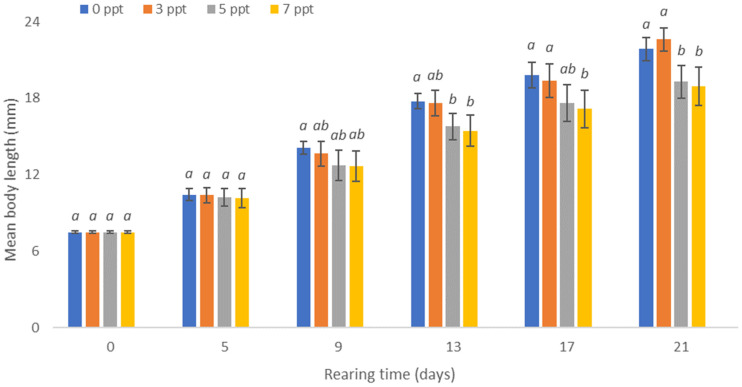
Increase in the body length of the blue bream *Ballerus ballerus* (L.) larvae reared in freshwater and in water with salinity of 3, 5, and 7 ppt. The same letter indexes above the columns in a given sample mean that the values are not statistically significantly different (α = 0.05).

**Figure 4 animals-13-01245-f004:**
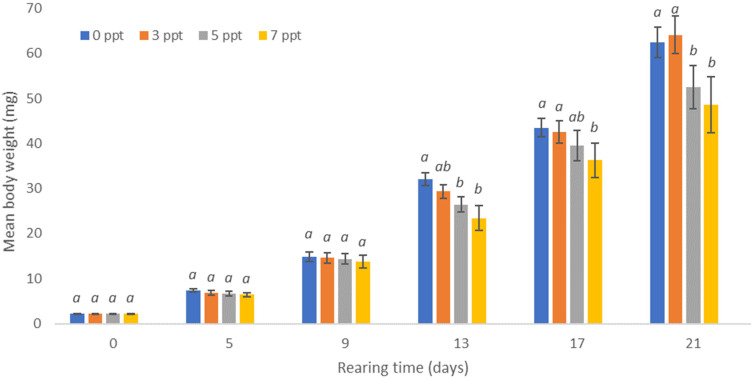
Increase in the body weight of blue bream *Ballerus ballerus* (L.) larvae reared in freshwater and in water with salinity of 3, 5, and 7 ppt. The same letter indexes above the columns in a given sample mean that the values are not statistically significantly different (α = 0.05).

**Table 1 animals-13-01245-t001:** Results of rearing the blue bream *Ballerus ballerus* (L.) larvae in freshwater and in water with salinity of 3, 5, and 7 ppt. Mean value ± SD. Results in rows with the same letter index are not statistically significantly different (α = 0.05).

	0	3	5	7
Initial mean body length (mm)	7.5 ± 0.1 ^a^	7.5 ± 0.1 ^a^	7.5 ± 0.1 ^a^	7.5 ± 0.1 ^a^
Initial mean body weight (mg)	2.2 ± 0.1 ^a^	2.2 ± 0.1 ^a^	2.2 ± 0.1 ^a^	2.2 ± 0.1 ^a^
Final mean body length (mm)	21.9 ± 0.9 ^a^	22.6 ± 0.9 ^a^	19.3 ± 1.3 ^b^	18.9 ± 1.5 ^b^
Final mean body weight (mg)	62.5 ± 3.4 ^a^	64.1 ± 4.2 ^a^	52.4 ± 4.8 ^b^	48.6 ± 6.2 ^b^
Initial stock (indiv.)	1000	1000	1000	1000
Final stock (indiv.)	986 ± 1.0 ^a^	970 ± 2.0 ^a^	946 ± 7.2 ^b^	832 ± 15.4 ^c^
Survival (%)	98.6 ± 0.1 ^a^	97 ± 0.2 ^a^	94.6 ± 0.7 ^b^	83.2 ± 1.5 ^c^
Average weight gain of larvae (PM) (mg)	60.3 ± 3.4 ^a^	61.9 ± 4.2 ^a^	50.2 ± 4.8 ^b^	46.4 ± 6.2 ^b^
Increase in total length (ITL) (mm·d^−1^)	0.7 ± 0.0 ^a^	0.6 ± 0.1 ^a^	0.6 ± 0.1 ^b^	0.5 ± 0.1 ^b^
Relative growth rate (RGR) for weight (%·d^−1^)	17.3 ± 0.3 ^a^	17.4 ± 0.4 ^a^	16.3 ± 0.5 ^b^	15.9 ± 0.7 ^b^
Relative growth rate (RGR) for length (%·d^−1^)	5.4 ± 0.2 ^a^	5.4 ± 0.3 ^a^	4.6 ± 0.3 ^b^	4.5 ± 0.4 ^b^
Relative growth rate (RBR) for biomass (RBR) (%·d^−1^)	17.2 ± 0.3 ^a^	17.3 ± 0.4 ^a^	16.0 ± 0.5 ^b^	14.9 ± 0.7 ^b^
Biomass (g·dm^−3^)	2.5 ± 0.1 ^a^	2.5 ± 0.2 ^a^	2.0 ± 0.2 ^b^	1.6 ± 0.2 ^b^
Biomass (g)	61.6 ± 3.4 ^a^	62.2 ± 4.1 ^a^	49.6 ± 4.5 ^b^	40.4 ± 5.2 ^b^

**Table 2 animals-13-01245-t002:** Relative mean final length (RFL), weight (RFW), and biomass (RFB) of experimental fish were calculated assuming that final length, weight, and biomass of the control 0 ppt were 100%.

Salinity (ppt)	RFL (%)	RFW (%)	RFB (%)
0	100.00	100.00	100.00
3	103.4	102.6	101.0
5	88.2	83.9	80.5
7	86.7	77.7	65.6

## Data Availability

The data presented in this study are available on request from the corresponding author.

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
