# Peer review of "Effect of Different Water Salinities on the Larvae of the Blue Bream Ballerus ballerus (Linnaeus, 1758) during Rearing"

_animals, 2023, doi:10.3390/ani13071245_

Round 1

Reviewer 1 Report

The article “Effect of Different Water Salinities on The Larvae of The Blue

Bream Ballerus ballerus (Linnaeus, 1758) During Rearing” is devoted to the influence of salinity on biological indicators of fish.

 While reading the article, I’ve got some questions and comments:

1. What water was used as a control?

2. What water was used to prepare test solutions of 3, 5, 7 ppt?

3. Tab. 1 and in the text - all values ​​​​of the measured indicators must be given

to the first decimal place

4. Tab. 1 - why was the Final Stock given with SD, if the experiment was done in one repetition?

5. Fig. 2, 3 it is necessary to explain what letters a, b and their combination above the columns means

6. As a recommendation, you can give a photo of the experimental setup. How were the measurements of length and weight without larvae damage?

 After correcting the text, taking into account the comments, the article can be published.

Author Response

Dear Reviewer,

Thank you very much for your insightful review. All comments were taken during review process marked in the text. All main changes were marked in the text in blue.

The article “Effect of Different Water Salinities on The Larvae of The Blue Bream Ballerus ballerus (Linnaeus, 1758) During Rearing” is devoted to the influence of salinity on biological indicators of fish. While reading the article, I’ve got some questions and comments:

  1. What water was used as a control?

Tap water without salinity was used in the experiment.

  1. What water was used to prepare test solutions of 3, 5, 7 ppt?

Tap water was also used to prepare suitable water salinity.

  1. 1 and in the text - all values ​​​​of the measured indicators must be given to the first decimal place

As suggested by the Reviewer, all numerical values have been corrected, both in the tables and in the text to one decimal place.

  1. 1 - why was the Final Stock given with SD, if the experiment was done in one repetition?

Due to the Editor’s suggestion to shorten the Materials and Methods section, we did not notice that we mistakenly removed the sentence stating that the experiment was performed in three

repetitions. The sentence about the number of repetitions was added in the text.

  1. 2, 3 it is necessary to explain what letters a, b and their combination above the columns means

The same letter indexes above the columns in a given sample means that the values are not

statistically significantly different (p ≤ 0.05). It was added to the text.

  1. As a recommendation, you can give a photo of the experimental setup. How were the measurements of length and weight without larvae damage?

Before taking the measurements, the captured larvae were sedated in a solution of MS-222 (tricaine methanesulphonate) (50 mg·l-1 of water). Next, they were weighed on an analytical balance KERN ALJ 220-5 DNM with accuracy up to 0.1 mg (KERN & Sohn GmbH, Ballingen, Germany). Larvae still anaesthetised were placed with some amount of water on Petri plates and observed under a stereoscopic microscope LEICA MZ16Z (Leica Microsystems GmbH, Wetzlar, Germany). The size of larvae was analysed in LAS V 3.1.0. software (Leica Microsystems GmbH, Wetzlar, Germany). The total body length (longitudo totalis - l. t.) was measured with the accuracy of 0.1 mm. After these treatments, the larvae were placed in a container filled with well-oxygenated water for recovery. After recuperation from anaesthesia and starting to swim actively, the larvae were returned to the aquaria from which they had been taken.

The Materials and Methods section has been shortened on the request of the Editorial Board. We have also used a reference to an earlier manuscript (Influence of Water Salinity on the Growth and Survivability of Asp Larvae Leuciscus aspius (Linnaeus, 1758) under Controlled Conditions), where the Methodology is described in more detail. An illustrative circuit diagram created for this experiment has been added to the Materials and Methods section.

After correcting the text, taking into account the comments, the article can be published.

Reviewer 2 Report

Review report for the manuscript “Effect of Different Water Salinities on The Larvae of The Blue Bream Ballerus ballerus (Linnaeus, 1758) During Rearing

The study reports on the effects of 3 different salinity levels on growth and survival rate of a freshwater fish species under climate change perspectives. Overall, the MS is well-written and interesting to read. However, there are some important areas in the MS that deserve improvements. I would recommend this MS for publication with the following suggested corrections.

Results:

·     Table 1:  Many data are not necessary. For example, data of initial length, weight, stock density can be briefly introduced in the experimental set up (methodology section). Increase in total length (ITL) (mm·d-1)” or index of total length gain per unit time (ITL) should be “absolute growth rate or specific growth rate).

·     I would suggest presenting only the standard growth rate data as below for table 1.  

-        Mean length and weight gained

-        Absolute growth rate (AGR) or specific growth rate (SGR) for fish length and weight

-        Relative growth rate (RGR) for fish length and weight

-        Final biomass gained

·             The survival rate should be clearly incorporated with mortality rate at the beginning of the result section.

Other minor type errors:

Line 44: content, no dash –

Line 191-192: Font size and colour of the sentence need to be consistent with entire MS.

Line 227: The study confirms that blue bream larvae, like asp larvae asp larvae …” delete “asp larvae”.

Line 174-175: On the last day of rearing, the blue bream larvae reared in 3 ppt water gained an average body weight of 64.1 mg with an average length of 22.6 mm (Figure 2 and Figure 3)?

Line 187-188: “These were differences that were not statistically significant” should be “These differences were not statistically significant

Author Response

Dear Reviewer,

Thank you very much for your insightful review. All comments were taken during review process marked in the text. All main changes were marked in the text in blue.

Review report for the manuscript “Effect of Different Water Salinities on The Larvae of The Blue Bream Ballerus ballerus (Linnaeus, 1758) During Rearing

The study reports on the effects of 3 different salinity levels on growth and survival rate of a freshwater fish species under climate change perspectives. Overall, the MS is well-written and interesting to read. However, there are some important areas in the MS that deserve improvements. I would recommend this MS for publication with the following suggested corrections.

Results:

Table 1:  Many data are not necessary. For example, data of initial length, weight, stock density can be briefly introduced in the experimental set up (methodology section). Increase in total length (ITL) (mm·d-1)” or index of total length gain per unit time (ITL) should be “absolute growth rate or specific growth rate). I would suggest presenting only the standard growth rate data as below for table 1.  

The formulae used to calculate the parameters in the table have been removed from the paper after the suggestion of the Editor, leaving only a reference to our previous publications. We believe that the data contained in Table 1 are more readable and understandable to the reader, so we would like to leave them as they are. The parameters are commonly used in scientific publications related to the rearing of fish larvae under different conditions. The second Reviewer had no objection to the parameters included in the table. He only suggested rounding the numbers to one decimal place. If the Reviewer feels that it is necessary to remove some of the data this will be done.

-        Mean length and weight gained

-        Absolute growth rate (AGR) or specific growth rate (SGR) for fish length and weight

-        Relative growth rate (RGR) for fish length and weight

-        Final biomass gained

The survival rate should be clearly incorporated with mortality rate at the beginning of the result section.

 As suggested by the reviewer it was been corrected.

Other minor type errors:

 Line 44: content, no dash –

It was corrected.

Line 191-192: Font size and colour of the sentence need to be consistent with entire MS.

It was corrected.

Line 227: The study confirms that blue bream larvae, like asp larvae asp larvae …” delete “asp larvae”.

It has been removed from the text.

Line 174-175: On the last day of rearing, the blue bream larvae reared in 3 ppt water gained an average body weight of 64.1 mg with an average length of 22.6 mm (Figure 2 and Figure 3)?

It was corrected.

Line 187-188: “These were differences that were not statistically significant” should be “These differences were not statistically significant”

It was corrected.